# Virtual Reality and Exercise Training Enhance Brain, Cognitive, and Physical Health in Older Adults with Mild Cognitive Impairment

**DOI:** 10.3390/ijerph192013300

**Published:** 2022-10-15

**Authors:** Ja-Gyeong Yang, Ngeemasara Thapa, Hye-Jin Park, Seongryu Bae, Kyung Won Park, Jong-Hwan Park, Hyuntae Park

**Affiliations:** 1Department of Health Sciences, Graduate School, Dong-A University, Busan 49315, Korea; 2Laboratory of Smart Healthcare, Dong-A University, Busan 49315, Korea; 3Department of Neurology, College of Medicine, Dong-A University, Busan 49201, Korea; 4Health Convergence Medicine Laboratory, Biomedical Research Institute, Pusan National University Hospital, Busan 49241, Korea

**Keywords:** virtual reality, exercise, mild cognitive impairment, cognition, electroencephalogram

## Abstract

We investigated the effectiveness of virtual-reality-based cognitive training (VRCT) and exercise on the brain, cognitive, physical and activity of older adults with mild cognitive impairment (MCI). Methods: This study included 99 participants (70.8 ± 5.4) with MCI in the VRCT, exercise, and control groups. The VRCT consisted of a series of games targeting different brain functions such as executive function, memory, and attention. Twenty-four sessions of VRCT (three days/week) were performed, and each session was 100 min long. Exercise intervention consisted of aerobic and resistance trainings performed in 24 sessions for 60 min (2 times/week for 12 weeks). Global cognitive function was measured using the Mini-Mental State Examination (MMSE) test. Resting-state electroencephalography (EEG) of the neural oscillatory activity in different frequency bands was performed. Physical function was measured using handgrip strength (HGS) and gait speed. Results: After the intervention period, VRCT significantly improved the MMSE scores (*p* < 0.05), and the exercise group had significantly improved HGS and MMSE scores (*p* < 0.05) compared to baseline. One-way analysis of variance (ANOVA) of resting-state EEG showed a decreased theta/beta power ratio (TBR) (*p* < 0.05) in the central region of the brain in the exercise group compared to the control group. Although not statistically significant, the VRCT group also showed a decreased TBR compared to the control group. The analysis of covariance (ANCOVA) test showed a significant decrease in theta band power in the VRCT group compared to the exercise group and a decrease in delta/alpha ratio in the exercise group compared to the VRCT group. Conclusion: Our findings suggest that VRCT and exercise training enhances brain, cognitive, and physical health in older adults with MCI. Further studies with a larger population sample to identify the effect of VRCT in combination with exercise training are required to yield peak benefits for patients with MCI.

## 1. Introduction

Mild cognitive impairment (MCI), defined as a decline in cognitive function higher than that expected for an individual’s age [1], is considered a prodromal stage of dementia. Individuals with MCI have an increased risk of developing dementia with a conversion rate of 46% over a period of three years [2]. Over time, some individuals with MCI seem to remain stable or revert to normal cognition [1], and intervention at this stage may help patients with MCI to do so. Hence, MCI can be considered an ideal period at which preventive intervention strategies can be applied to help sustain functioning in MCI patients, which may delay progression to a clinical diagnosis of Alzheimer’s dementia.

A growing number of studies have focused on pharmacological and non-pharmacological strategies that improve brain and cognitive health in older adults [3]. Among the non-pharmacological interventions, increasing evidence suggests that exercise prevents cognitive decline and impairment. With over a decade of evidence on the benefits of exercise on cognition, the recently updated practice guidelines of the American Academy of Neurology (AAN) on MCI included exercise as a level B recommendation as an approach to improve cognition function [4] in individuals with MCI. In a review of observational studies involving 1254 participants aged 65 years and over, it was reported that exercise provides protection against MCI [5]. Similarly, exercise intervention studies on MCI patients [6,7,8] and randomized controlled trials (RCTs) have also established the cognitive benefits of exercise [9,10,11]. Several RCTs of older adults with and without MCI have found that exercise has a considerable positive effect on memory and executive functions [12,13,14,15], brain structure [12,14], and brain function [16]. Despite the proven benefits of exercise, it is often considered difficult to perform and people tend to prefer entertaining leisure activities instead [17]. This leads to a lack of proper approaches that enhance motivation and adherence, which are crucial factors in achieving the benefits of exercise intervention.

The extent to which an individual is motivated and adheres to their exercise regimen can affect their outcomes. This is because those with better adherence have been reported to be more likely to achieve better physical fitness and performance, as well as higher self-perceived effects of exercise [18]. A study conducted in community-dwelling older adults with MCI showed that higher adherence to exercise sessions leads to better cognitive performance [19,20]. Despite these positive findings, there remain concerns that older adults with MCI or dementia are physically inactive; hence, their adherence to exercise after the intervention is poor [15]. Therefore, alternative interventions are required to overcome this drawback. Virtual reality (VR) is one of these alternatives, which has currently gained high popularity in the research field of MCI and dementia [21]. VR, a computer-simulated environment, provides an artificial interactive environment closely representing reality [22], providing the user with a feeling of virtually “being there”. VR has several features, one of which is that it can be adjusted according to the individuals’ needs and their performance in different activities and tests [22], leading users to feel a sense of control and enjoyment. This could potentially increase motivation and adherence to the intervention.

VR interventions used for cognitive rehabilitation have resulted in improvements in cognition (e.g., memory, dual tasking, processing, and attention) [23], and physical [24] and psychological functioning [25]. Likewise, in addition to exercise, cognitive training (CT) is another popular intervention for MCI. An RCT comparing cognitive, exercise, and VR training has reported that VR is the most favored by older adults without cognitive impairment [26]. In our previous study, we performed a virtual-reality-based cognitive training (VRCT) on individuals with MCI, and our results showed improvements in cognition, brain function, and physical performance [27]. However, at present, a head-to-head comparison of the effectiveness of VR-based and/or exercise interventions is lacking; hence, a definite conclusion cannot be reached. Exploring the differences in and effectiveness of these interventions side-by-side may lead to useful insight into which intervention is the most beneficial. Therefore, in the present study, we aimed to explore and compare the brain, cognitive, and physical benefits of VRCT and structured exercise interventions in older adults with MCI. Furthermore, we explored the effect of both interventions on brain function measured with resting-state electroencephalography (EEG).

## 2. Materials and Methods

### 2.1. Subject

This randomized controlled trial included participants recruited from regional cohorts in Busan Metropolitan city, Korea. The inclusion criteria for the study were: (i) participants must be ≥55 and <85 years old, (ii) participants must be able to perform physical activity without aid, and (iii) participants must be diagnosed with MCI based on medical evaluations consisting of neurological examinations and detailed neuropsychological assessments conducted by a dementia specialist. Sample size was calculated using G*Power 3.1.9.4 [28]. Previous studies using VR and/or exercise intervention with similar outcome variables (i.e., global cognitive function and EEG band power) were used to estimate the sample size [29,30]. We derived a small-to-medium effect size of intervention from these studies (Cohen’s f = 0.29). Thus, with a statistical power of 0.80, a probability level of 0.05, and an effect size of 0.29, a sample size of 90 participants was deemed necessary to achieve sufficient power. Considering 10% dropout, we recruited 99 participants and randomly allocated them into three groups (*n* = 33 in each group): (i) VRCT intervention, (ii) exercise intervention and (iii) control. Figure 1 describes the study design, with participants’ recruitment and exclusion criteria. The study procedures were approved by the University Institutional Review Board (IRB No. 2-1040709-AB-N-01-201909-HR-036-0 and 2-1040709-AB-N-01-201901-HR-006-04). All the participants signed an informed consent form prior to enrollment in the research study.

### 2.2. VRCT and Exercise Intervention

The VRCT training period was performed in total of 24 sessions. The participants attended 24 sessions of the VRCT, thrice a week, with each session lasting 100 min. The session consisted of three 20 min sessions of VRCT and four 10 min session of eye massage. The VR training consisted of four games that targeted different domains of the brain, such as attention and working memory. The game contents are described in detail in previous study [27]. An Oculus VR headset (Oculus quest headset) and two wireless hand controllers (left and right hands) were used for VR training.

The exercise intervention included a warm-up, an aerobic and resistance training period, and a cool-down. Exercise training was performed in 24 session (2 times/week, 12 weeks). Aerobic exercise intensity was monitored at a moderate intensity using Borg’s Rating of Perceived Exertion (RPE) [31] and a target heart rate. Participants performed indoor and outdoor walking, and jumping jack, skip jump step box walking. The protocol for resistance training programs was progressive in loading (2–3 set and 65% of 1 repetition maximum). Free weight (weight-bearing) and/or Thera-band were used to provide the training stimulus. The muscular endurance training stimulus was subsequently increased using the 10–15 RM method: 2 sets of from 10 to 15 repetitions were completed with proper form and without discomfort. Resistance training were targeted the six large muscle groups: hamstrings, quadriceps, gastrocnemius, biceps, triceps, erector spinae. All the exercises were performed with the instructions and under the supervision of a fitness instructor.

The control group attended eight sessions on education seminars on health-related topics (nutrition and exercise tips on prevention of geriatric disease). Each session lasted for 30 min.

### 2.3. EEG Recording, Preprocessing and Analysis

Brain activity was measured using a wireless dry electroencephalogram (EEG) headset (Cognionics Inc., San Diego, CA, USA). The electrodes (Fp1, Fp2, F7, F3, Fz, F4, F8, T3, C3, Cz, C4, T4, T5, P3, Pz, P4, T6, O1 and O2) were placed according to the built-in design of the device which was based on the international 10–20 position system. The EEG data were acquired at a sampling rate of 500 Hz and were filtered through a high-band-pass filter (0.5 Hz) and low-band-pass filter (120 Hz). Electrode impedance was maintained at >5 kΩ. EEG data acquisition was performed in a dimly lit, quiet room, with the participants in a resting state. Their eyes were closed for 5 min.

EEG noise preprocessing and analyses were conducted using the iSyncBrain v.2.1.0, 2018 (iMediSync Inc., Seoul, Korea). The EEG data underwent band-pass filtering, with the frequency ranging between 1 and 45 Hz section. Additionally, a 60 Hz notch filter was used to remove noise from the power supplies. Subsequently, the common average reference was applied to remove the noise mixed throughout the recorded EEG data. Artifacts were filtered and removed using bad epoch rejection and independent component analysis to generate clean data for the subsequent analysis. At the sensor level, using the fast Fourier transform (FFT) analysis, the power spectral density of the cleaned EEG data was computed and decomposed into the following frequency bands, as shown in Table 1.

The power ratio was calculated by dividing the low-frequency power (delta and theta) by the high-frequency power (alpha and beta). The analyzed power ratio index are as follows:Theta/beta (TBR): This tends to reflect attention-related functions. Increased TBR is a predictor of poor cognitive and attention control [33].Theta/alpha (TAR): This reflects cognitive ability, especially learning and memory-related functions. Increased TAR is associated with decreased cognitive ability [34].Delta/alpha (DAR): This is associated with cognitive deficit. Increased DAR is associated with cognitive impairment [35].

In the source-level analysis, cortical activity across the brain was assessed using the standardized low-resolution brain electromagnetic tomography technique (sLORETA), to compare relative power values in regions of interests (ROIs) and connectivity between ROIs. We used imaginary coherence (iCoh) as a measure of functional connectivity because it was an excellent measure of brain interactions. Furthermore, it had minimal volume conduction problems.

### 2.4. Cognitive Function and Anthropometric Measures

Global cognitive function was measured using the Mini-Mental-State Examination (MMSE) test. The electronic version of the trail-making test (TMT) A, and symbol digit substitution test (SDST), developed by the National Center for Geriatrics and Gerontology Functional Assessment Tool (NCGG-FAT) were used to assess cognitive function [36]. In the TMT-A, the participants were required to connect the numbers scattered on a page in consecutive orders by drawing a line. The time taken to complete TMT was recorded in second (s), with a maximum period of 90 s. In the SDST, the participants were required to match the pairs of symbols to their corresponding numbers. During the test, the target symbol was displayed, and the participants were required to match this with the number corresponding to it. One point was awarded for each correctly matched number. The time limit for SDST was set at 90 s.

Anthropometric measurements, such as height, weight, and body mass index (BMI), were measured, and socio-demographic measures such as age, sex, and education were also acquired.

### 2.5. Physical Function

Hand grip strength (HGS) was measured using a hand dynamometer (TKK 5101 Grip-D Takei co. Inc., Tokyo, Japan). The measurement was performed in a standing position on a flat surface, and participants were asked to keep their arms slightly apart from their bodies and grab the dynamometer with all their strength while pointing towards the ground during the test. They were then instructed to maximally clench the hand to hold this. The procedure was performed twice for the dominant hand, with a minute rest between tests. The average of the two measurements (in kilograms) was then recorded.

The gait speed test was performed over a 7 m stretch, at a usual pace, which included an acceleration phase of 1.5 m and a 4 m walk, followed by a deceleration phase of 1.5 m. A 4 m distance was measured and marked the starting and ending points on the floor. At the start of the measurement, the participants were verbally instructed to walk at a comfortable pace. The timed up-and-go (TUG) test was used to assess mobility. The participants performed specific sequences of movements, such as getting up from a chair with a height of 43 cm with a back support, walking a distance of 3 m, turning around, walking back to the chair, and sitting again. The test was performed twice, and the shortest time (in seconds) was used for analysis.

Appendicular skeletal muscle mass (ASM) was measured using multi-frequency segmental Bioelectrical impedance analysis (BIA) to assess whole-body and appendicular fat mass and lean soft tissue (InBody S10, Inbody Co., Ltd., Seoul, Korea). The appendicular skeletal muscle mass index (SMI) was calculated by adding the fat-free lean mass of all four limbs, which was then normalized by the square of height. Participants with metal implants or pacemakers were excluded from the BIA. Measurements were taken before the commencement of physical tests and as per manufacturer’s manual. The participants were standing in an upright position, with their feet and hands in contact with the base and grip electrodes, respectively.

### 2.6. Statistical Analysis

Statistical analyses were performed using the IBM SPSS Statistics, version 26.0, 2021 software package for Windows (SPSS Inc., Chicago, IL, USA). The Shapiro–Wilk test was used to determine the normality of the data distribution. Participant characteristics and baseline performance were compared among the three groups using One-Way ANOVA for normally distributed interval variables, Kruskal–Wallis tests for ordinal and non-normal interval variables. Since we were also interested in the effect of time on the outcome measures, we performed mixed group × time repeated-measures analysis of covariance (ANCOVA) after adjusting sex, age, and years of education. In case of statistically significant group × time interactions, group-specific post hoc tests were calculated to identify the statistically significant comparisons. For the preprocessed EEG data, sex- and age-adjusted ANCOVA was performed to assess the differences in power ratio and whole brain connectivity between the VRCT and exercise groups. Statistical significance was set at *p* <  0.05.

## 3. Results

The baseline demographic characteristics and physical and cognitive functions of all participants are described in Table 2.

As shown in Table 3, the MMSE score significantly increased in both the VRCT, and exercise groups compared to baseline (*p* < 0.05). Significant positive changes were also observed in the TMT-A in the VRCT (*p* < 0.05) during follow-up compared to baseline. The SDST score at follow-up was higher in the VRCT group than in the exercise group (*p* < 0.05).

### 3.1. EEG

#### 3.1.1. Band Power: Theta [4–8 Hz]

Theta occurs in a deeply relaxed state of mind. This is important for processing information and creating memories [37]. A significant decrease in theta power was observed in the parietal area (*p* = 0.036) during follow-up in the VRCT group compared with that in the exercise group as shown in Figure 2.

#### 3.1.2. Power Ratio

(i)Theta Beta ratio (TBR)

The ratio of theta (4–7 Hz) to beta oscillations (13–30 Hz), is known as the theta–beta ratio (TBR). This was calculated by dividing the power density in theta by that in beta. One ANOVA analysis of post-intervention measurements showed that the VRCT and exercise groups had lower TBR ratios after the intervention compared to the control group as shown in Figure 3. The post-hoc comparison further showed significant reductions in TBR in the exercise group compared to that in the control group (*p* < 0.0187).

(ii)Delta Alpha ratio (DAR)

Delta wave powers are increased in patients with MCI due to loss of cortico-cortical connectivity [38], which, in turn, is caused by damage in pathways with dense cholinergic fibers [39]. Moreover, a decrease in alpha wave power may be associated with cholinergic dysfunction, which is observed in MCI [40]. The ratio of delta (1–4 Hz) to alpha oscillations (8–12 Hz), known as the delta–alpha ratio (DAR), was calculated by dividing the power density in delta by that in alpha. A significant decrease in the DAR was observed in the frontal (*p* < 0.01) and temporal regions (*p* < 0.03) of the brain in the exercise group compared to VRCT group post-intervention as shown in Figure 4.

#### 3.1.3. Resting-State Whole-Brain Connectivity

The differences in resting-state alpha connectivity were compared between the VRCT and exercise groups. Higher connectivity was observed in the prefrontal cortex, anterior cingulate cortex, and temporal and parietal areas of the brain in the exercise group than in VR group (Figure 5).

### 3.2. Physical Functions

Figure 6 shows a significant improvement in the VRCT and exercise groups compared with those in the control group. The skeletal muscle mass of the appendages (ASM) only significantly increased in the exercise group, but there was a slight, not significant, increase in the VRCT group. In addition, the walking speed was significantly decreased in the control group. However, it was maintained and increased in VRCT and exercise groups.

## 4. Discussion

The present study aimed to investigate whether virtual-reality-based exercise training had a beneficial effect on improving brain, cognitive, and physical functions in MCI patients, to compare its effect with that of the control educational program group. The results revealed that virtual-reality-based training improved cognitive and brain function in MCI patients compared to the use of an exercise program. However, the exercise program showed a greater improvement in physical function than the VRCT group. Although exercise is a widely used intervention, with proven benefits of MCI to prevent dementia and delay cognitive decline [14,41], VRCT is currently gaining popularity in dementia research. However, evidence regarding its benefit on MCI patients is sparse compared to evidence on exercise intervention [27]. The present study aimed to compare the effectiveness of exercise training and VRCT on cognitive and brain health in older patients with MCI. We found that both exercise and VRCT intervention improved global cognitive function and showed positive changes in different brain regions. In the exercise group, positive changes were observed in the frontal, temporal, parietal, and central brain regions, whereas, in the VRCT group, changes were observed in the parietal region of the brain.

In MCI patients, beta and theta band powers are the first to change [42,43]. Theta waves are increased because of the slower axonal conduction time in the subcortical area [44]. Slowing is caused by damage to cholinergic neurons in the lateral capsular and Perisylvian pathways, which play important roles in cortical activity. They significantly contribute to cortical activity [45]. Furthermore, they are rich in cholinergic fibers [39]. We measured the EEG-related power changes in the brain. Our findings showed a decrease in the theta band power in the VRCT in the parietal brain area. This area of the brain is said to be actively involved in creating the feeling of presence in VR applications [46,47]. The superior parietal lobe participates in visual imagery [48], and mental transformations of the body-in-space [49]. On the other hand, the angular gyrus (part of the inferior parietal lobe) was engaged in the encoding and retrieval of schema-associated memories [50]. This explains the positive changes observed in the parietal region of the brain in the VRCT group. The decrease in theta band power could be indication of neuronal level changes in the brain due to the VRCT intervention.

We also measured TBR and DAR power as EEG indices as sources of critical information about brain activity. The increase in TBR power has been reported to be observed during mind wandering [51]. Mind wandering [52] is a phenomenon wherein thoughts are not controlled by top-down processes or voluntary allocation of attention. It is a predictor of poor executive cognitive control [53,54]. Similarly, increased DAR power, particularly in the frontal area, has been associated with cognitive impairment [36]. In our study, we observed a significant decrease in TBR and DAR power in the exercise group compared to the VRCT and control groups. Exercise increases the levels of major cholinergic neurotransmitters, such as acetylcholine [55]. It has been reported that, after exercise, there is a constant increase in BDNF levels [56], which induces neurogenesis in the hippocampus. Thus, exercise improves overall brain health. In our study, the changes in EEG indices reflect these improvements. Another point of note in our study was that a decrease in TBR power was observed in the central area, which included the temporal and parietal lobes, whereas a decrease in DAR power was observed in the frontal, temporal, and parietal brain areas. Aerobic exercise has been reported to increase attentional processing and plasticity in the frontal, central, and parietal midline brain areas. It has also been associated with increased glucose metabolism in the temporal lobe [36], reflecting higher brain activity. This explains the positive effect of exercise intervention on certain brain areas in the frontal, central, parietal, and temporal regions in our study.

The final measured EEG index in our study was whole-brain alpha connectivity. Alpha waves are predominantly present during the resting state of the brain [57]. They originate from the thalamo-cortical and cortico-cortical interactions that are modulated by neurotransmitters, such as acetylcholine [58]. Reduced alpha connectivity indicates the disruption of the inhibitory function of alpha waves [59] due to the pathology of neurodegenerative diseases. A recent meta-analysis on the resting-state alpha connectivity and cognition has reported that patients with MCI have reduced alpha connectivity [60]. In our study, we observed a higher connectivity in the exercise group after the intervention period when comparing the differences in resting-state alpha connectivity. Moreover, brain areas with higher connectivity were the prefrontal cortex, posterior cingulate cortex, and medial temporal regions, which were considered part of the default mode network (DMN) of the brain. There is evidence of reduced DMN in patients with MCI [61]. These changes in brain function due to exercise and VRCT exercise might have led to an improvement in cognitive function in those groups, as reflected by the MMSE scores. Both the exercise and VRCT groups showed a significant increase in the MMSE score compared to the baseline. Additionally, in the VRCT group, the TMT-A was significantly enhanced. We measured the effect of exercise training and VRCT on physical function as a secondary outcome.

We observed a significant increase in HGS and TUG scores in both the exercise training and VRCT groups in the post-intervention measurements. Hand grip strength (HGS) is a simple and cost-effective method of evaluating the overall skeletal muscle strength and quality, while the Timed Up-and-Go test (TUG) is a simple test used to assess a person’s mobility. It requires both static and dynamic balance. Despite these age-related effects on muscle and mobility, VRCT and exercise interventions were found to increase strength and mobility. The increase in these physical functions in the intervention group may be due to enhanced muscle strength, balance, and mobility. Conventionally, resistance exercise training is prescribed to induce skeletal muscle growth, and growing evidence suggests that exercise training or game-based virtual-reality training can also induce strength and balance ability [62]. Exercise training in our study included a combination of aerobic and resistance exercise, which explained the improved HGS. In the VRCT group, a similar result was observed, even though the intervention targeted cognition. In young adults, leap motion controller (LMC) VR therapy has been reported to increase grip strength and balance [63]. A LMC is a commercially available device that is used to improve a user’s hand function. Although we did not use LMC in our study, during the VRCT, the participants had to use hand-held controllers and performed upper-limb movements to perform their intended activities, which might have led to positive improvement in the HGS.

Despite the promising results, one limitation of the present study was the difference in the frequency of intervention. There were a total of 24 intervention sessions, but the intervention frequency and duration were different. While the exercise group performed training twice a week for 12 consecutive weeks, the VRCT was performed three times a week for eight weeks. Secondly, in our study, participants were predominantly female. A third limitation of this study was the small sample size and the length of the intervention. Consequently, further longitudinal studies in large population samples are needed to clarify that the combination of VRCT and exercise training provides the best benefits in older adults with MCI.

## 5. Conclusions

In summary, our findings indicate that both VRCT and exercise intervention independently improve the physical, cognitive, and brain health of older adults with MCI. VRCT, however, had a greater benefit on cognition, whereas exercise training had a greater effect on physical function in this population. Additionally, we found that these interventions enhanced different domains of the brain. Exercise training had a positive effect on the frontal, temporal, and parietal brain regions, whereas VRCT had a positive effect on the parietal region. We believe that by combining exercise training and VRCT, greater cognitive benefits will be achieved, since the intervention effects can be expanded to larger brain areas whilst improving physical health.

## Figures and Tables

**Figure 1 ijerph-19-13300-f001:**
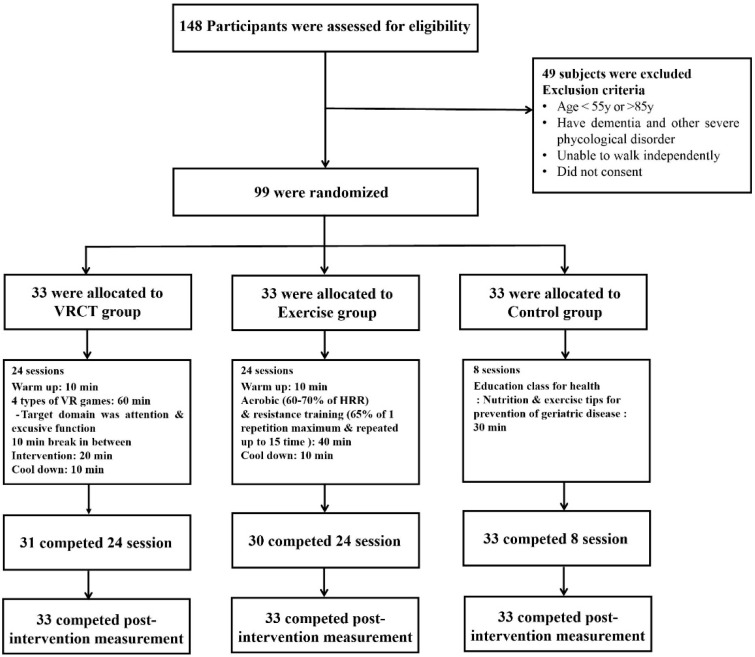
Flow diagram of participants throughout intervention.

**Figure 2 ijerph-19-13300-f002:**
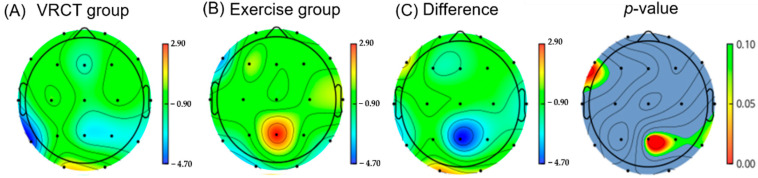
This shows the difference in Theta band power in the VRCT group compared to the exercise group at follow-up. (The color bars (**A**–**C**) indicate power density: blue indicates lower power density and red indicates higher power density. The color bar next to *p*-value indicates the *p* value ranging from 0.10 (green) to 0.00 (red).

**Figure 3 ijerph-19-13300-f003:**
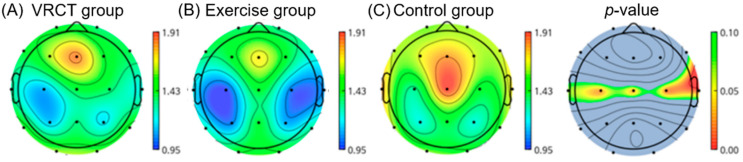
This is a figure shows difference in theta/beta power ratio (TBR) in the central area (parietal region) of the brain in exercise group, compared to VRCT and control group at post-intervention. (The color bars (**A**–**C**) indicate power density: blue indicates lower power density and red indicates higher power density. The color bar next to *p*-value indicates the *p* value ranging from 0.10 (green) to 0.00 (red).

**Figure 4 ijerph-19-13300-f004:**
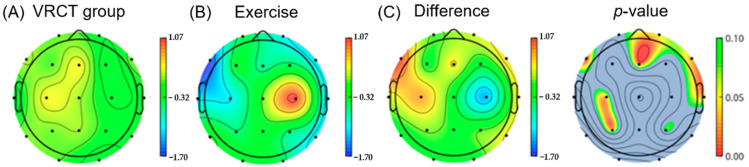
This shows the difference in delta/alpha power ratio (DAR) in the exercise group compared to the VR group at follow-up. The exercise group showed a significant decrease in DAR in the frontal and temporal regions of the brain compared to the VR group. (The color bars (**A**–**C**) indicate power density: blue indicates lower power density and red indicates higher power density. The color bar next to *p*-value indicates the *p* value ranging from 0.10 (green) to 0.00 (red).

**Figure 5 ijerph-19-13300-f005:**
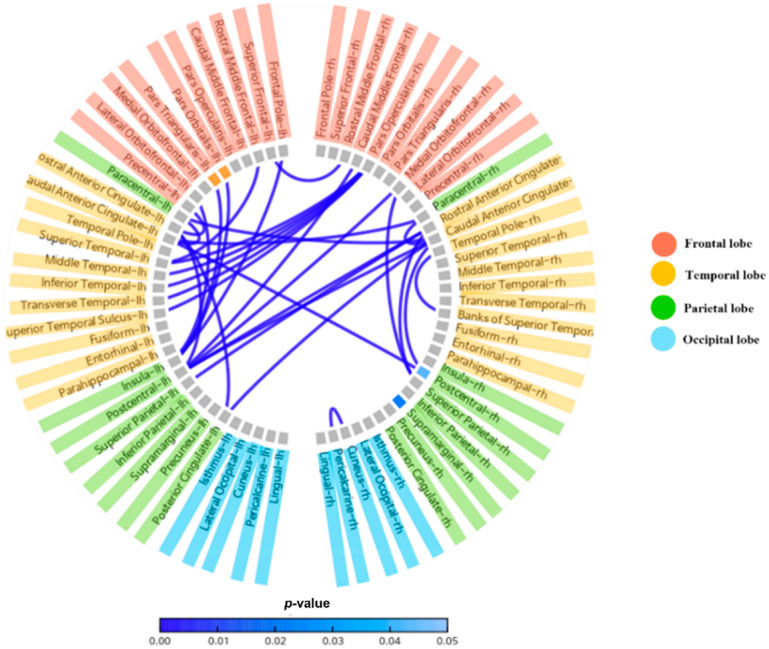
This figure shows *p*-value of the lower alpha rhythm (8~10 Hz) connectivity of different brain regions.

**Figure 6 ijerph-19-13300-f006:**
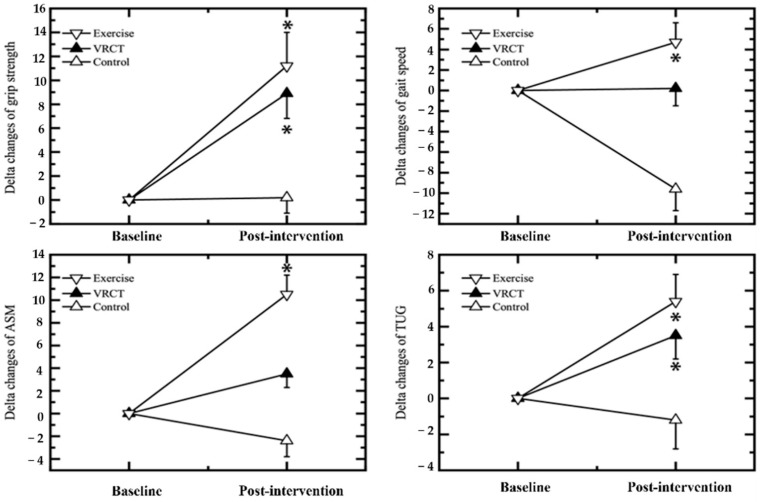
Percent changes in the physical functions in the VRCT, exercise and control groups between baseline and post-intervention. Values are expressed means ± SE; * indicates significant differences between the control groups (Bonferroni correction test, *p* < 0.05). VRCT, virtual-reality-based cognitive training; ASM, appendicular skeletal muscle; TUG, timed up-and-go test, * represents a significant difference (*p* < 0.05) intervention group vs. control group.

**Table 1 ijerph-19-13300-t001:** Brain frequency bands and their general characteristics.

Frequency Band	Frequency (Hz)	General Characteristics of Frequency Band
Delta	1–4	Sleep
Theta	4–8	Deeply relaxed, inward focused
Alpha	8–12	Very relaxed, passive attention
Beta	12–30	Anxiety dominant, active, external attention, relaxed
Gamma	30–45	Concentration

This table is adapted and modified from Abang et al. [32].

**Table 2 ijerph-19-13300-t002:** The baseline anthropometric, physical, and cognitive function characteristics of the participants.

Variables	VRCT	Exercise	Control
Number (female/male)	33 (20/13)	33 (30/3)	33 (27/6)
Age (years)	72.5 ± 5	67.9 ± 3.6	72.6± 5.6
Education (years)	9.5 ± 3.7	8.5 ± 3.9	8.5 ± 3.6
Height (m)	1.58 ± 0.1	1.55 ± 0.4	1.57 ± 0.1
Weight (kg)	61.9 ± 8.8	58.9 ± 7.8	61.5 ± 9.8
BMI (kg/m^2^)	24.6 ± 2.7	24.5 ± 3.2	24.7 ± 3.2
SBP (mmHg)	128.9 ± 18	131.9 ± 19	130.3 ± 14
DBP (mmHg)	69.1 ± 12	81.1 ± 11	74.6 ± 11
Grip Strength (kg)	24.3 ± 6.6	21.2 ± 4.34	22.9 ± 5.9
Gait speed (m/s)	1.10 ± 0.19	1.14 ± 0.18	1.12 ± 0.21
ASM (kg)	20.1 ± 6.1	19.3 ± 5.7	18.3 ± 5.3
TUG (s)	11.2 ± 1.8	10.9 ± 1.7	10.5 ± 2.2
MMSE (score)	27.21 ± 1.9	26.9 ± 1.7	26.5 ± 2.8
TMT-A (s)	42.7 ± 19.5	44.7 ± 17.9	44.7 ± 20.6
SDST (score)	39.6 ± 15.4	37.4 ± 13.3	30.2 ± 12.4

Data are expressed in mean ± standard deviation. BMI: Body Mass Index; SBP: Systolic Blood Pressure; DBP: Diastolic Blood Pressure; ASM: Appendicular skeletal muscle mass; TUG: Timed Up-and-Go Test; MMSE: Mini-Mental State Examination, TMT: Trail-Making Test, SDST: Symbol Digit Substitution Test.

**Table 3 ijerph-19-13300-t003:** The comparison of global cognitive function between baseline and post-intervention the Control, VR and Exercise intervention group.

Variables	VRCT		Exercise		Control	
	Baseline	Post-Intervention	Baseline	Post-Intervention	Baseline	Post-Intervention
MMSE (score)	27.2 ± 1.9	28.1 ± 1.7 *+	26.9 ± 1.7	27.8 ± 1.6 *	26.5 ± 2.8	26.7 ± 1.9 +
TMT-A(s)	42.7 ± 19.5	36.4 ± 18.1 *	44.7 ± 17.9	42.4 ± 20.1	44.7 ± 20.6	46.1 ± 22.1
SDST (score)	39.6 ± 15.4	53.2 ± 13.7 *+	37.4 ± 13.3	49.4 ± 12.9 *+	30.2 ± 12.4	32.1 ± 13.4 +

Data are expressed in mean ± standard deviation. Mixed group × time repeated-measures analysis of covariance (ANCOVA) for each outcome, adjusted for age, gender, and education; * represents a significant difference (*p* < 0.05) between baseline and post-intervention within the group; + represents a significant difference (*p* < 0.05) in post-intervention between groups; MMSE: Mini-Mental State Examination, TMT: Trail Making Test, SDST: Symbol Digit Substitution Test.

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
