# Peer review of "Virtual Reality and Exercise Training Enhance Brain, Cognitive, and Physical Health in Older Adults with Mild Cognitive Impairment"

_ijerph, 2022, doi:10.3390/ijerph192013300_

Round 1

Reviewer 1 Report

This is an informative article directly comparing the effectiveness of VR-based interventions and exercise intervention. This research adds to the body of evidence that VR can improve cognitive function, especially in adults with MCI. It also brings more specificity to the respective benefits of VR and exercise. I have the following suggestions:

1.      In general, this article would benefit from more explanation on alpha, beta, delta, and theta waves, including their characteristics and functions. It would help the audience appreciate a lot more the significance of the wave powers and ratios and what their changes signify. For example, the authors describe the theta wave as “It occurs in deeply relaxed state of mind important for processing information and making memories”. This statement gives the impression that theta wave is related to critical brain functions and therefore you want theta power to be higher. However, the rest of the article indicates otherwise – a decrease in theta power is associated with positive changes. More context around the waves and ratios can help clarify the audience’ understanding and resolve this kind of disconnect.

2.      The Abstract is well written; however, the rest of the article is filled with minor grammatical errors and typos. For example, “people tend to a have preference for entertaining leisure activities types” should be “people tend to have a preference for entertaining leisure activities types”; “The extent to which an individual motivates and adhere to their exercise regimen can affect their outcomes…” can be changed to “The extent to which an individual is motivated and adheres to their exercise regimen can affect their outcomes…”; “A study conducted on community-dwelling older adults with MCI showed higher adherence to exercise sessions that have better cognitive performance” can be corrected to “A study conducted in community-dwelling older adults with MCI showed those with higher adherence to exercise sessions have better cognitive performance”.  In “The VR training consisted of four games that targets different domain of brin such as attention and working memory”, “brin” should be “brain” and "targets" should be "target". I suggest carefully reviewing the article and having it edited by a professional editor who is a native English speaker.

3.      From Table 1, it seems like only males are included in this study. Is that true? The Methods and Results sections should clearly identify the gender of the participants.

4.      A 99-participant study seems more like a Phase II trial. Is there plan to conduct a larger scale, Phase III trial to substantiate the study results?

5.      In the first row of Table 1 where the variable is n (male), what do the numbers in and outside the parentheses mean, respectively? For example, what does 27 and 6 mean in “27 (6)”?

6.      The authors describe Table 2 as “The comparison of physical function and global cognitive function between baseline and follow up in the Control, VR and Exercise intervention group”. All 3 measures in Table 2 are for cognitive functions though. How can the authors justify its claim on physical function based on data in Table 2?

7.      For Fig. 2, 3 and 4, do the numbers on the color bars indicate fold changes? It is not clear from the article. Also, does the color in the circles indicate the change from baseline to follow-up in each group?

Author Response

Reviewer 1 :Comments and Suggestions for Authors

This is an informative article directly comparing the effectiveness of VR-based interventions and exercise intervention. This research adds to the body of evidence that VR can improve cognitive function, especially in adults with MCI. It also brings more specificity to the respective benefits of VR and exercise. I have the following suggestions:

1.In general, this article would benefit from more explanation on alpha, beta, delta, and theta waves, including their characteristics and functions. It would help the audience appreciate a lot more the significance of the wave powers and ratios and what their changes signify. For example, the authors describe the theta wave as “It occurs in deeply relaxed state of mind important for processing information and making memories”. This statement gives the impression that theta wave is related to critical brain functions and therefore you want theta power to be higher. However, the rest of the article indicates otherwise – a decrease in theta power is associated with positive changes. More context around the waves and ratios can help clarify the audience’ understanding and resolve this kind of disconnect.

       Thank you for your appropriate comment and your suggestion is extremely insightful. As your suggestions, in order to better reach the audience with clarity, we have added the following information for the brain frequency bands (delta, theta, alpha beta) and power ratios in method section as follows:

Line 210: Table1

Line 217-223:

  1. Theta/beta (TBR) : It tends to reflect attention-related functions. Increased TBR is a predictor of poor cognitive and attention control. [30]
  2. Theta/alpha (TAR): It reflects cognitive ability, especially learning and memory-related functions. Increased TAR is associated with decreased cognitive ability. [31]

iii. Delta/alpha (DAR): It is associated with cognitive deficit Increased DAR is associated with cognitive impairment. [32]

  1. The Abstract is well written; however, the rest of the article is filled with minor grammatical errors and typos. For example, “people tend to a have preference for entertaining leisure activities types” should be “people tend to have a preference for entertaining leisure activities types”; “The extent to which an individual motivates and adhere to their exercise regimen can affect their outcomes…” can be changed to “The extent to which an individual is motivated and adheres to their exercise regimen can affect their outcomes…”; “A study conducted on community-dwelling older adults with MCI showed higher adherence to exercise sessions that have better cognitive performance” can be corrected to “A study conducted in community-dwelling older adults with MCI showed those with higher adherence to exercise sessions have better cognitive performance”. In “The VR training consisted of four games that targets different domain of brin such as attention and working memory”, “brin” should be “brain” and "targets" should be "target". I suggest carefully reviewing the article and having it edited by a professional editor who is a native English speaker.

     We apologize for proof-reading the manuscript well. Thank you for your comment. We re-checked and a native English speaker thoroughly reviewed our manuscript for all typos, spellings, prepositions, punctuation, and grammar.

  1. From Table 1, it seems like only males are included in this study. Is that true? The Methods and Results sections should clearly identify the gender of the participants.

     We apologize for any misunderstanding. 22 men and 77 women participated in our study. Table 2 (previously table1) has been revised as follows:

  1. A 99-participant study seems more like a Phase II trial. Is there plan to conduct a larger scale, Phase III trial to substantiate the study results?

      Thank you for your comment. As we have described in the limitation, we understand your curiosity, therefore we are planning to conduct further longitudinal studies in large population samples to identify relationship between exercise and VRCT with large sample and to investigate the effect of combination intervention in older adults with MCI.

  1. In the first row of Table 1 where the variable is n (male), what do the numbers in and outside the parentheses mean, respectively? For example, what does 27 and 6 mean in “27 (6)”?

     We apologize for the insufficient description. The outside the parentheses indicates the total number of participants in each group, and the number inside the parentheses indicates the number of female and male. Duplicates our response to comment 3, we have corrected the notation in table 2 (previously table1).

  1. The authors describe Table 2 as “The comparison of physical function and global cognitive function between Baseline and follow up in the Control, VR and Exercise intervention group”. All 3 measures in Table 2 are for cognitive functions though. How can the authors justify its claim on physical function based on data in Table 2?

        We apologize for the misunderstanding caused by using words that don’t describe the table. Thank-you for kindly pointing this issue. The inclusion of “physical function” in the table description is a mistype on our part. We have changed the description as follows:

Line 300-301: The comparison of global cognitive function between Baseline and post-intervention in the Control, VR and Exercise intervention group.

  1. For Fig. 2, 3 and 4, do the numbers on the color bars indicate fold changes? It is not clear from the article. Also, does the color in the circles indicate the change from Baseline to follow-up in each group?

     Thank-you for your comment. As your suggestion, we have added description of the color changes to provide further clarification to the readers in the figure legends. The description added is as follows in figure 3, 4 and 5 (previously figure 2, 3, and 4):  The color bars indicate power density: blue indicate lower power density and red indicate higher power density. The right color bar next to p-value indicate the p value ranging from 0.01(green) to 0.00 (red).

Reviewer 2 Report

The abbreviation should be spelt out in its first use e.g., VRCT in the introduction

What was the design of the study?

How was the sample size calculated?

More information is required about the sample recruitment. From where were samples recruited? What were the inclusion and exclusion criteria?

Please add a diagram for the participants’ flow

The exercise intervention written is incomplete. Please provide detailed information about the intervention provided. Which resistance exercise was performed? Targeting which muscles?

Was the data normally distributed?

The result and discussion is adequate 

Author Response

Reviewer 2

  1. The abbreviation should be spelt out in its first use e.g., VRCT in the introduction

     Thank you for the comment. We apologize for this mistake. Thank you for pointing it out. We have made the necessary changes to the manuscript.

  1. What was the design of the study?

      Thank you for the comment. The study was a randomized controlled trials. We added the information about our study design in the method section of the manuscript as follows:

Line 115-116: This randomized controlled trial included participants recruited from regional cohorts in Busan Metropolitan city, Korea…

  1. How was the sample size calculated?

     Thank you for the comment. We have included the process of sample size calculation in the method section of the manuscript as follows:

Line 116-118: Using a power of 80%, a significance level of 0.05, and an effect size of 0.2, the necessary sample size was assessed to be 99 participants.

  1. More information is required about the sample recruitment. From where were samples recruited? What were the inclusion and exclusion criteria?

    Thank you for your comments and suggestions. We apologize for not providing enough information about the recruitment, inclusion, and exclusion criteria. We have expanded the information on sample recruitment and inclusion criteria in method section as follows:

Line 115-122: This randomized controlled trial included participants recruited from regional cohorts in Busan Metropolitan city, Korea. Using a power of 80%, a significance level of 0.05, and an effect size of 0.2, the necessary sample size was assessed to be 99 participants. The inclusion criteria for the study were: i) participants must be 55 years old, ii) must be able to perform physical activity without assistance, and iii) be diagnosed with MCI based on medical evaluations consisting of neurological examinations and detailed neuropsychological assessments conducted by a dementia specialist.

The exclusion criteria and further information on sample recruitment has been described in a flow chart added in the method section as figure1.

  1. Please add a diagram for the participants’ flow. (figure1)

 Thank you for the suggestion. We have added the flowchart below to the manuscript under the subjects in the method section as mentioned in comment 4.

The exercise intervention written is incomplete. Please provide detailed information about the intervention provided. Which resistance exercise was performed? Targeting which muscles?

    Thank you for your comments. We have added a detailed description of the exercise training performed as follows:

Line 151-165:  The exercise intervention included a warm-up, an aerobic and resistance training period, and a cool-down. Exercise training was performed in 24 session (2 times/week, 12 weeks). Aerobic exercise intensity was monitored at a moderate intensity using Borg's Rating of Perceived Exertion (RPE) [28] and a target heart rate. Participants performed indoor and outdoor walking, and jumping jack, skip jump step box walking. The protocol for resistance training programs was progressive in loading (2-3 set and 65% of 1 repetition maximum). Free weight (weight-bearing) and/or Thera-band were used to provide the training stimulus. The muscular endurance training stimulus was subsequently increased using the 10-15 RM method—2 sets of 10 to 15 repetitions were completed with proper form and without discomfort. Resistance training were targeted the six large muscles group: hamstrings, quadriceps, gastrocnemius, biceps, triceps, erector spinae. All the exercises were performed with the instructions and un-der the supervision of a fitness instructor.

  1. Was the data normally distributed?

      Thank you for the comment. We have added information regarding the normality of the data, and the type of analysis used in case of non-normal distribution in the statistical analysis section as follows: 

Line 265-269: The Shapiro–Wilk test was used to determine the normality of the data distribution. Participant characteristics and baseline performance were compared among the three groups using One-Way ANOVAs for normally distributed interval variables, Kruskal-Wallis tests for ordinal and non-normal interval variables…..

  1. The result and discussion are adequate.

     Thank you very much for your kind comments.

Round 2

Reviewer 2 Report

The sample size calculation is not clear. More details are required on how you calculated the sample size. Which data was used?

Author Response

The sample size calculation is not clear. More details are required on how you calculated the sample size. Which data was used?

     Thank you again for the comment that allowed us to improve the quality of the manuscript. We agree with your suggestion, and we added the information for sample size calculation in the manuscript accordingly.

 Line 120-127: Sample size was calculated using G*Power 3.1.9.4 [28]. Previous studies using in the VR and/or exercise intervention with similar outcome variables (i.e., global cognitive function and EEG band power) were used to estimate the sample size [29, 30]. We derived a small-to-medium effect size of intervention from these studies (Cohen’s f= 0.29). Thus, with a statistical power of 0.80, a probability level of 0.05, and an effect size of 0.29, sample size of 90 participants were deemed necessary to achieve sufficient power. Considering 10% dropout, we recruited 99 participants and randomly allocated them into three groups (n=33 in each group):